# A Design Framework for Contextual and Embedded Information Visualizations in Spatial Augmented Reality

Nikhita Joshi*1    Matthew Lakier†1    Daniel Vogel‡    Jian Zhao§

Cheriton School of Computer Science, University of Waterloo

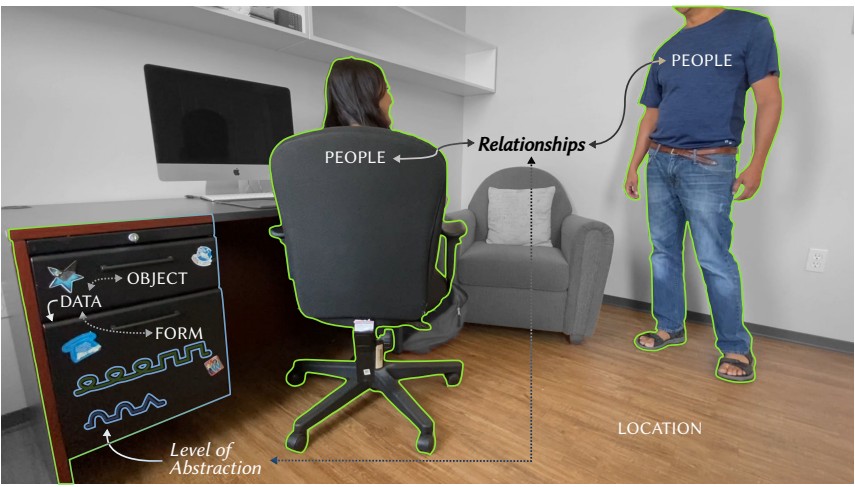

Figure 1: Demonstration of how different factors and properties in our SAR InfoVis design framework can be represented, overlaid on one of our exemplar applications (`office-cabinet`).

## ABSTRACT

Spatial augmented reality (SAR) displays digital content in a real environment in ways that are situated, peripheral, and viewable by multiple people. These capabilities change how embedded information visualizations are used, designed, and experienced. But a comprehensive design framework that captures the specific characteristics and parameters relevant to SAR is missing. We contribute a new design framework for leveraging context and surfaces in the environment for SAR visualizations. An accompanying design process shows how designers can apply the framework to generate and describe SAR visualizations. The framework captures how the user's intent, interaction, and six environmental and visualization factors can influence SAR visualizations. The potential of this design framework is illustrated through eighteen exemplar application scenarios and accompanying envisionment videos.

## 1 INTRODUCTION

With recent trends like personalized self-tracking and Internet of Things (IoT) devices, data is becoming more tightly coupled to the real, physical world. Situated [39], embedded, and physicalized visualizations [41] enable a deeper integration of data in the physical world by placing information near, or on objects that are associated with the underlying data. Advances in augmented reality (AR) have made it easier to create situated and embedded visualizations using mobile phones [40] and head-mounted displays (HMDs) [1, 5, 39].

---

*e-mail: nvjoshi@uwaterloo.ca

†e-mail: matthew.lakier@uwaterloo.ca

‡e-mail: dvogel@uwaterloo.ca

§e-mail: jianzhao@uwaterloo.ca

1The first two authors contributed equally to the work.

Unlike conventional see-through or HMD-based AR, spatial augmented reality (SAR) is a type of AR that places digital content into the physical world using projectors [30], allowing any surface within an environment to become an information display without peering through a phone or wearing an HMD. Using compact projector-camera units [20, 38] and steerable displays [28, 42], this technology could become as common as the lightbulb in future home and office environments. SAR creates several new opportunities for information visualizations (InfoVis): (1) it is always available, which could help make AR more pervasive [8]; (2) it does not require people to wear or shift their attention to another device; (3) it delivers the same AR experience to multiple people simultaneously; and (4) it can avoid the privacy concern of continuously sensing the environment using a camera.

Within an environment, there are many unused surfaces and objects with unique physical properties that could be leveraged to create situated, embedded, and physicalized visualizations with SAR. The pervasiveness of SAR means the primary mode of use for visualizations is context-driven [8], and there are many environmental properties that could be used, like those proposed by Schmidt et al. [35]. However, it is unclear how this context should be combined with other aspects, like the user's intent, implicit and explicit interaction, available objects and surfaces, underlying data, and the visual form [41] to create meaningful visualizations. General visualization design frameworks [3, 21, 22], alternative visualization pipelines [14, 41], and frameworks for pervasive or embedded visualizations through AR [1, 8] do not capture all characteristics and parameters relevant to SAR. A SAR-specific framework is necessary to address unique challenges, like sharing and coordinating views between people, and providing ambient awareness through peripheral displays embedded in the environment. Previous frameworks consider context as limited to just people or locations, or their goal is to only inform the visualization's underlying data.

We contribute a design framework for embedded InfoVis in SAR. The framework captures the user's intent, interaction, and six envi-

ronmental and visualization factors represented in a Venn diagram: PEOPLE, LOCATION, TIME, OBJECT, DATA, and FORM (Figure 1). Each factor has its own set of properties and each can influence other factors. Unlike HMD-based AR, which can display content mid-air and in personalized views, SAR must be placed on objects and content has to be shared publicly. Our SAR framework emphasizes *objects* by placing it at the centre of the Venn diagram, signifying its importance to both the environment and the visualization. An accompanying design process shows how the framework can be used to generate new visualization designs and describe existing designs in previous work.

To illustrate how this design framework can be used for SAR visualizations, we generate eighteen exemplar applications to show visualizations used throughout a typical day at home and in the office. We use envisionment videos created with video post-production and special effects software to test the framework and demonstrate the range of visualizations and contexts with which it can be used.[1] As classified by Munzner [24], our design framework could be considered a "model", and the construction of our exemplar applications could be considered a form of "design study" involving demonstrating how designers can use different aspects of the design framework. Our design framework has two key benefits: it provides descriptive power, allowing designers to discuss SAR visualization characteristics using a common language, and it generates future InfoVis use cases and applications by guiding design decisions.

## 2 BACKGROUND AND RELATED WORK

Our work is related to situated, embedded, and physicalized visualizations, SAR visualizations, and InfoVis design frameworks.

### 2.1 Situated, Embedded, & Physicalized Visualizations

AR has made it easier to place visualizations on objects in the environment. *Situated visualizations* are placed in physical locations that are strongly associated with the underlying data, bridging the spatial and semantic gap between visualizations and their underlying data [39]. Previous work has focused on creating situated visualizations using *see-through* AR, in other words, AR using mobile phones or tablets. For example, SiteLens [39] is a hand-held device to show information about air quality at specific locations; and White et al. [40] developed a mobile application to display information about plants using a video see-through view.

*Embedded visualizations* are a type of situated visualization where the visualization is directly placed on top of relevant objects in the environment, rather than simply in close proximity [41]. For example, ElSayed et al. [5] create an HMD AR application for displaying grocery data directly on the product packaging, and the RFIG Lamp system [29] uses handheld SAR to render information about the contents of a box directly onto the box surface. *Embedded physicalizations* are visualizations designed to "blend" in with objects, so that physical attributes, such as object shapes, affect how the visualization appears. For example, the product shelves in a store could change colour to indicate the sales performance of each product [41].

Like conventional AR, SAR encourages visualizations that are more situated and embedded in the physical world. SAR also enables physicalizations that take advantage of existing environment geometry. However, there are many aspects that may affect how SAR visualizations should be created. First, with HMD-based AR, everyone has a unique view of the augmented world, but SAR visualizations must be shared by multiple people, which may infringe on someone's privacy. Second, while conventional AR allows for "floating" content, this is not feasible with SAR, so content must be mapped onto physical surfaces. As SAR is always on, there is a need to create visualizations that "match" the space, so they are not

---

[1]The full set of envisionment videos is in the supplementary materials and can be viewed online at sarinfovis.github.io.

too distracting in everyday scenarios. Finally, conventional AR is intentional in nature, since a device must be used to see data, which can limit discovery. SAR's permanence in the environment and its ability to create physicalized visualizations could increase discovery.

### 2.2 InfoVis in SAR

SAR uses projectors to place content in the environment without the use of HMDs [30]. Previous work has largely focused on creating enabling technology for SAR. Ceiling-mounted steerable projectors like Everywhere Displays [28] and Beamatron [42], and projector-camera units, like Lightform [20] and I/O bulb [38], allow many surfaces within a room to be used as displays. SAR toolkits, like RoomAlive [15], automatically calibrate and localize projector-depth camera units, which simplifies the creation of SAR environments. Other work has focused on using SAR to support and enhance different activities, like gaming [15, 16], teleconferencing [27], and training physical tasks [37].

While there are situated or embedded visualizations that use conventional AR, few use SAR. One notable example is Raskar et al.'s RFIG Lamp system [29]. A handheld projector "reveals" metadata associated with boxes in a warehouse directly on the boxes, like lines showing a box's original position on a shelf if it was recently moved, and highlighting corners that have become damaged. Kane et al.'s Bonfire system [18] extends laptop-based interaction to a nearby desk surface using pico-projectors mounted to the sides of a laptop, a kind of small portable SAR. A demo application shows information about objects placed on a table; for example, a bar chart near a cup indicates coffee consumption. Xiao et al.'s WorldKit [43] uses projectors and depth sensors to track the user's hand in the environment, allowing them to "paint" interactive displays on various surfaces and objects. Their focus is on the enabling system, but they demonstrate a SAR visualization in the form of a circle that grows in size and changes colour as people place ingredients for a recipe on a table. SAR has also been used to visualize physiological data in the environment. Frey et al. show 3D topographic graphs of brain signal data on the head of a figurine [7]. As part of a board game, they also project infographics and histograms about heart rate on player boards to promote social interactions [6].

Previous work suggests that SAR can be used to create a wide range of visualizations, but there still lacks a systematic way to describe, understand, and design SAR visualizations.

### 2.3 InfoVis Design Frameworks

We define a design framework as a conceptual model that captures relationships between key factors to consider for design. Our framework can be understood through the lens of Brehmer and Munzner's typology of visualization tasks [3]. The typology describes visualization tasks using the interrogative words "Why," "How," and "What." Because SAR visualizations are situated or embedded, they can present contextually-relevant information rather than being centred around supporting predefined analytic tasks. For this reason, our framework focuses on the "consume" tasks from Brehmer and Munzner's typology: "presenting," "discovering," and "enjoying."

Our framework also shares similarities with and draws ideas from general design models in information visualization. The Design Activity Framework (DAF) [21] guides designers through the process of developing a visualization, and it breaks down the process into four "activities": "understand," "ideate," "make," and "deploy". Our framework spans across the "understand," "ideate," and "make" stages of the DAF. Two key terminologies we borrow from the DAF are "constraints," or limitations that the designer must work with, and "considerations," criteria or softer limits that the designer should prioritize. Specifically, in each stage of our process using the framework, we identify applicable constraints and considerations. Sedlmair et al.'s nine-stage framework [36] offers a design study methodology from the initial stage of learning about visualizations to

the final stage of writing a design study paper. We see our framework as useful to designers during the "design" stage of the nine-stage framework, in which visual encodings, interaction, data abstraction, and so on, are designed. The Nested Blocks and Guidelines Model (NBGM) [22] extends Munzner's earlier nested model [25]. It uses four "levels": "domain," "abstraction," "technique," and "algorithm." The arrows in our framework diagram relate to the "between-level" guidelines of the NBGM, in that they suggest dependencies for how a designer might use one factor to inform the design of another. Our factors also relate to the blocks of the NBGM, in that some of our factors represent categories of specific blocks. For example, User Intent in our framework encompasses the NBGM's situation and task blocks, and we share the concepts of "data" and "visual encoding" blocks with the NBGM, as data theme and form "visual encoding" factor properties, respectively.

Bach et al. [1] develop the AR-CANVAS framework for designing HMD-based embedded AR data visualizations. They describe parameters such as the data from the environment, objects, the person viewing the visualization, the visualization itself, interactions, and the space. Our work is also factor-based and shares some similar elements. However, we adapt and extend several aspects necessary to apply it to SAR. First, AR-CANVAS considers the importance of context, such as the people in the environment, activities, and objects, but it uses a data-centric perspective, meaning that context is only used to infer what underlying data should be shown. We also integrate other ways in which the context can affect a visualization into our framework; for example, how the activity or people in the room affect the level of abstraction. Second, AR-CANVAS considers the person viewing the visualization in terms of physical aspects, such as their field of view, viewing direction, and distance from the visualization. Our work considers an expanded set of characteristics related to people, such as their personal interests, occupation, what they are doing, who they are with, and their relationships. AR-CANVAS notes the possibility of multiple people viewing the same visualization, but the nature of HMD-based AR means the framework relies on individual AR views to enable personal views. This differs from SAR, so our framework captures how visualizations should adapt to support multiple simultaneous viewers (e.g. moving to a different surface or enabling a "privacy mode"). We also extend Bach et al.'s description of how visualizations can be anchored to different objects (e.g. beside, on, around), by considering that some visualizations cannot be shown on certain objects (e.g. it is not practical to project on a television screen that is in use) and that visualizations must be projected on existing surfaces and cannot be rendered in mid-air. Finally, Bach et al. use mock-ups of a book library to realize their ideas. Our exemplar applications and envisionment videos include a wider variety of locations, activities, and people, data, objects, and visual forms.

## 3 DESIGN FRAMEWORK

Inspired by past work on situated, embedded, and physicalized visualizations, as well as previous InfoVis design frameworks, we developed a design framework for SAR InfoVis. It captures key factors and associated properties of the environment and visualization that can influence the design of a SAR visualization (Figure 2).

### 3.1 Framework Methodology and Scope

An iterative, *divergent-convergent* [12] methodology was used to generate our framework and corresponding design process. Concepts from previous work were synthesized, including Meyer et al.'s Nested Blocks and Guidelines Model [22], McKenna et al.'s Design Activity Framework [21], AR-CANVAS [1], work on ambient displays [9, 13, 23, 31], work on ubiquitous computing [8, 34, 35], and work on situated and embedded visualizations [39, 41]. Using these concepts as a base, a set of vocabulary related to context, SAR,

and InfoVis; like "activity," "owner," "texture," and "shape"; was compiled and used as *properties* for common FACTORS.

After identifying primary dependencies between factors, the overall set of relationships was organized as a Venn diagram with connected inner elements. The connections highlight a design process with three main stages: (1) user intent, (2) environment, and (3) visualization. Our focus is on creating visualizations in everyday environments that act primarily as information displays (i.e. "consume" tasks from Brehmer and Munzner's visualization typology [3]), as opposed to fully interactive visualization systems. We do not focus on integrating complex analytical tasks into the visualization designs. Another focus is on how contextual information and simple forms of input influence the content and form of a visualization, but we leave investigations of interaction techniques for future work.

### 3.2 Framework Description

The resulting framework diagram is illustrated in Figure 2 and its associated design process is described in Section 3.3. At a high level, our framework is composed of four aspects: the User Intent, Interaction, the Environment, and the Visualization. User Intent describes the goals, motivations, and desires of the people in the environment where the visualization appears. Consistent with previous InfoVis design frameworks and methodologies [22, 25, 36], we consider user information needs first. Interaction describes the ways in which the user can interact with the SAR visualization. This can be implicit or explicit, and can be achieved using many input modalities, such as voice, tangible, touch, or gestural input.

#### 3.2.1 Factors

There are six FACTORS representing key entities to consider when creating information visualizations in SAR: PEOPLE, LOCATION, TIME, OBJECT, DATA, and FORM (Table 1). Factors can be associated with the Environment (i.e. real world), the Visualization (i.e. projected content), or both. Factors related to the Environment define the context of the SAR visualization, which includes the location, nearby people and objects, and how they change over time [34]. Visualization factors define the visual characteristics of the projected SAR content, like where it will be placed, what data will be shown, and how the data will be represented visually. For example, visual representations that leverage existing shapes and textures in an environment can make a visualization appear more embedded [41]. A Venn diagram representation (Figure 2) is used to show which factors belong to the Environment, Visualization, or the intersection of both. Our framework places OBJECT at the centre of the Venn diagram. This placement signifies that it is both important as a form of context in the environment, and as a collection of surfaces on which to display a visualization, a duality which is unique to SAR.

#### 3.2.2 Properties

Each factor is defined by several *properties* that can influence and inform the design or effect of other factors and properties (represented as directional arrows in Figure 2). The ability for properties to influence one another is similar to the data-referent relationships proposed by Willett et al. [41], but we extend and formalize the possible effects that PEOPLE and the LOCATION may have on the DATA, OBJECTS, and FORM. Table 2 describes common and highly-influential factor properties, but this list is not exhaustive. As shown in previous work, like Schmidt et al.'s model for context-aware computing [35] and Grubert et al.'s taxonomy for pervasive AR [8], there are many other properties to consider for each factor. Our list serves to characterize the general types of properties associated with each factor.

#### 3.2.3 Influences Between Factors and Properties

Many properties are highly related to one another. Considering the Environment factors, many factor properties imply other factor

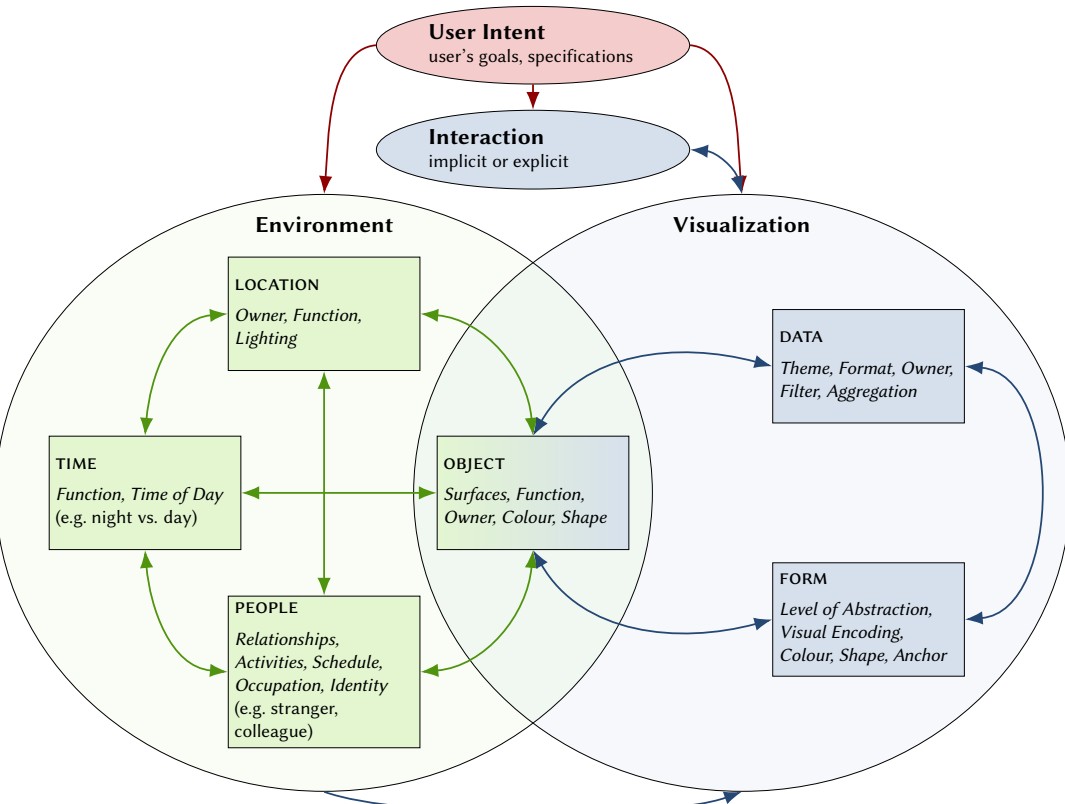

Figure 2: SAR design framework is modelled as a Venn diagram, encapsulating environmental and visualization factors (SMALLCAPS) and associated example properties (*italics*) that can influence the design of a SAR visualization. Arrow edges point towards the influenced factor and colour indicates the design stage (see Section 3.3).

properties, allowing designers to better understand the conditions in which the SAR visualization will exist. Schilit et al. [34] note that user actions and nearby objects can be inferred by context, such as the location, and these relationships are captured through the framework. The LOCATION and its *Function*, for example, may suggest nearby OBJECTS and what *Activities* PEOPLE are doing. Public LOCATIONS imply that the *Relationships* between PEOPLE are strangers, and that the OBJECTS have no *Owners*.

Considering the Visualization factors, the relationships between properties can make a SAR visualization appear more embedded in the environment [41]. DATA of a specific *Format* may be better represented as a certain FORM *Visual Encoding*, and this FORM *Visual Encoding* may have a unique *Shape*. Selecting OBJECT *Surfaces* with the same shape can make it appear more integrated into the environment, and the FORM'S *Colour* must contrast that of the OBJECT *Surface*, so the projected content can be seen.

Environmental properties can inform Visualization properties. For example, knowledge of *Personal Information* for PEOPLE, or LOCATION and OBJECT *Function*, could inform the DATA *Theme*, making the visualization more relevant to the person or the environment. Previous work on proximate selection suggests that nearby objects may be easier to use [34], so placing a visualization on an OBJECT *Surface* that is closer to the PEOPLE in a LOCATION may make it easier to view. FORMS with high *Levels of Abstraction* may be more easily viewed when PEOPLE are performing certain *Activities* that require the user's attention, like exercising. Privacy is an important consideration for SAR InfoVis, as visualizations are viewable by anyone in the same space. FORMS with high *Levels of Abstraction* may be more suitable for public LOCATIONS where strangers are likely present. Projecting at certain TIMES of day, like before bedtime, may be inappropriate, especially if these times are

typically associated with certain *Activities*, such as getting dressed. Certain factor properties in the Environment may change. Designers need to consider these ever changing Environment properties and how they could impact the design of Visualization factor properties.

## 3.3 Using the Associated Design Process

We used the framework descriptively as a lens for previous SAR visualizations, and generatively to create our exemplar applications, by following a three-stage process. This process is a combination of techniques used in previous work, including constraints and considerations from the DAF [21] and automatic contextual reconfiguration [34]. In the User Intent Stage, we consider the goals of the user, in the Environment Stage, we make inferences about the context, and in the Visualization Stage, we consider the design of the visualization and the ways that context and user interaction should inform and change it. The different arrow colours in Figure 2 indicate the different stages of the design process. We see this process as being applicable when designing a visualization, for example, in the "design" stage of Sedlmair et al.'s nine-stage framework [36]. We also use this design process as a lens to describe design decisions for existing visualizations in Section 4.1, and we outline how we used the framework to generate each exemplar application in Section 4.2.

Each stage involves identifying *constraints* and *considerations* that may impact the visualization design. We use similar definitions as McKenna et al.'s DAF [21]: constraints are limits placed on the visualization the designer must work with, like preserving privacy in public spaces, while considerations are looser restrictions that the designer should apply, and are usually associated with the visualization's aesthetics or usability. Each stage also involves identifying factors to be *fixed* and factors to be left *variable*, progressively refining the design of the visualization. Fixing a factor means specifying

Table 1: Description of Environmental and Visualization factors.

| Factor | Association | Definition |
|--------|-------------|------------|
| PEOPLE | Environment | Who is present in the environment that may be able to see or interact with the visualization. |
| LOCATION | Environment | Where the visualization will be shown; note this refers to a larger region, such as which room, rather than a specific projection surface within a room. |
| TIME | Environment | When the visualization will be shown. |
| OBJECT | Environment, Visualization | The specific surfaces in the environment, such as tables, walls, or shelves, on which the SAR visualization is placed. |
| DATA | Visualization | The underlying information that is conveyed by the visualization; note that this factor represents the dataset as a whole, rather that individual data entries. |
| FORM | Visualization | The visual form [41] of the resulting SAR visualization, including how the data is encoded, visualization colours, overall shape and size, and other visual characteristics. |

that it is context-independent and non-interactive once placed in the environment. For example, if a designer is creating a visualization to only be used in the bedroom, they could specify the LOCATION as fixed. In contrast, leaving a factor variable means specifying that it is context-dependent or interactive. For example, a visualization could hide personal details if a stranger comes into the room. Variable factors are similar to Schilit et al.'s notion of "automatic contextual reconfiguration" [34], in which components of a system change depending on contextual factors like the location or people present. In other cases, the method may lead to a factor being considered as not applicable. For example, a designer might decide that the time of day is not relevant to a visualization that shows the contents of a filing cabinet. We describe each stage in more detail and describe high-level guidelines that we developed through the process of creating our exemplar applications.

### 3.3.1 User Intent (red arrows)

For each visualization, we create a *user intent statement*, a sentence summarizing the goal the user intends to achieve. The user intent statement is typically centred around specific DATA that the user wants to visualize. For example, "I want to keep track of what's stored in my personal office cabinet." Other user intent statements are possible, such as wanting to see "something relevant" on a specific OBJECT. It can vary from vague to specific depending on the situation, but if it is vague, the statement should provide clues that can help a designer narrow down the visualization design decisions. For example, even though the above user intent statement does not mention a specific Environment LOCATION, because the OBJECT is a personal office cabinet, the designer could assume the LOCATION is an office environment. Designers can take more creative liberty when the user intent statement is vague. User intent statements can be elicited using existing methods that are common in UX research, such as interviews, focus groups, and field studies. At this stage, we consider any constraints or considerations, factors that are fixed, and factors that are variable, as indicated in the user intent statement. We also consider explicit interactions with the visualization; for example, the user may wish to filter or group time-series data by year, which would require some form of input (e.g. touch).

### 3.3.2 Environment (green arrows)

Similar to the first stage, we consider constraints and considerations, factors that can become fixed, and factors that will be left variable, but at this stage, we consider how we might fix different Environment factors based on what we already know. In other words, we make inferences and educated assumptions regarding more specific details about the context. This helps to narrow down the problem space for the visualization to address. Many considerations at this stage can draw from knowledge from ubiquitous computing. If the LOCATION is known to be "public" or "private" relative to the user, we can make assumptions about who will be present. For example, we may assume that a visualization in a bedroom is unlikely to be observed by strangers. Similarly, *Activities* are often associated with particular LOCATIONS or performed at particular TIMES, and vice versa. For example, if we create a visualization to support cooking, we might assume that the LOCATION is a kitchen, and that kitchen tools and appliances are present in the environment.

### 3.3.3 Visualization (blue arrows)

In this stage, we consider how the user intent and the previously-fixed Environment factors might inform the design of the visualization itself. We also make creative decisions, such as visualization colour schemes and designs. Our framework primarily focuses on how designers can integrate the proposed factors when designing a visualization, rather than the design of the form's visual encoding.

The LOCATION, *Activities* of PEOPLE, and DATA identified earlier often have strong associations with OBJECTS and can become fixed. Table 3 describes a number of examples of considerations for how designers may choose to design a visualization based on other factors. For example, a kitchen might contain a stove and be associated with the activity of cooking, and a stove will have data, like temperature, associated with it. The surfaces of objects also have to be considered at this stage. For example, projecting onto the screen of a TV that is turned on, or onto a tablecloth with complex embroidery, could obscure the visualization. Legends, labels, and scales may be omitted to better integrate into the environment as more embedded and aesthetically pleasing SAR visualizations. Related work from ambient displays suggests that this is suitable, as visualizations placed in the physical environment are lived with [9] and people can learn to interpret their meaning over time.

There are also a number of technological aspects to consider when using SAR systems in practice. For example, projectors need to be configured in the environment to cover useful areas on which to visualize, and also need to be mounted at angles that minimize distortion. Handheld projector systems would place further constraints on the size, visibility, and distortion of a SAR visualization. As SAR systems improve and become more commonplace, there will be fewer technological constraints on SAR visualizations.

We return to any visualization factors that we have not fixed by this point, to define how the visualization should respond in real-time to changes in context and through explicit user interactions. For example, at this stage, we might design various SAR buttons that the user can tap to hide or reveal labels. Privacy considerations are also important address in this stage; if the PEOPLE at a LOCATION have variable *Relationships*, the FORM'S *Level of Abstraction* could change as people enter and leave the environment.

## 4 APPLYING THE FRAMEWORK

We show how the framework and design process can be used to describe and generate SAR information visualizations. The motivation is threefold. First, applying the framework allowed us to illustrate how the framework could be used to generate and describe a variety of visualizations tailored to different environments. Second, describing existing examples allowed us to illustrate how the factors could be used independently from the design process. Third, generating new examples helped us iterate on the framework itself: as more

Table 2: Example properties for Environmental and Visualization factors.

| Property | Associated Factors | Definition | Example |
|---|---|---|---|
| *Function* | TIME, LOCATION, OBJECT | The primary way the factor is used. | Work activities usually occur within a set time frame in the day, a kitchen is primarily used when cooking, and a cabinet for storing small objects. |
| *Owner* | LOCATION, OBJECT, DATA | The individual or group to which a factor belongs, which may vary across factors. | Data about a single person's heart rate can be shown on a communal table. |
| *Lighting* | LOCATION | The amount and quality of ambient lighting. | A room could be lit with neutral-tone daylight or with the warm light of a lamp bulb. |
| *Relationships* | PEOPLE | How people are related. | People may be family, friends, colleagues, or strangers. |
| *Activity* | PEOPLE | What people are doing in a location. | People may be attending a meeting, reading, driving, or exercising. |
| *Personal Information* | PEOPLE | Other interesting and relevant information about people. | Other information may include personal schedules, occupations, or personal interests. |
| *Colour*, *Shape*, *Texture* | OBJECT | Describes the material properties of specific surfaces. | A table may be circular or rectangular and have a checkered tiled top, white laminate edges, and brown wood grain legs. |
| *Surfaces* | OBJECT | Describes different parts of individual objects. | A table has a tabletop, edges, and legs. |
| *Level of Abstraction* | FORM | How detailed the FORM should be. | Abstract patterns or ambient lighting would have high levels of abstraction, but detailed dashboards would not. |
| *Visual Encoding* (*Colour*, *Transparency*, *Shape*, *Size*, etc.) | FORM | How the data is represented visually. This could include many types of encodings, like *Colour*, *Transparency*, *Size*, or *Shape*, and each can be represented using their own FORM-specific properties. | FORM may be a line graph or bar chart. |
| *Anchor*, *Orientation* | FORM | Where the FORM is placed relative to the OBJECT. | A graph may be shown above, below, or beside an object when it cannot be used, like a television screen. |
| *Theme* | DATA | High-level topic or theme of the DATA from a user perspective. | Data may be sleep data, or work-related information. |
| *Filter*, *Aggregation* | DATA | Processes and functions that can be performed on the DATA. | Filter the data to show information related to the people in the room, or to only show averages from the past week. |
| *Format* | DATA | High-level classification of the DATA from an InfoVis perspective. | DATA may be time series data. |

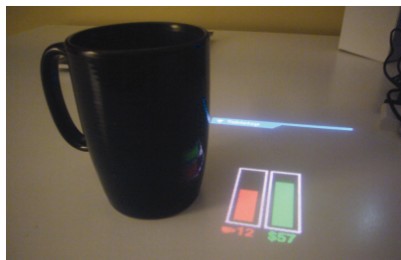

Figure 3: Coffee consumption visualization from Bonfire (Figure 5 of Kane et al. [18]).

applications were produced, new factor properties were identified, which allowed us to refine and validate the resulting diagram.

## 4.1 Describing Existing Visualizations

We use our framework and design process to describe existing SAR visualizations. We describe visualizations from two seminal SAR papers, Bonfire [18] and RFIG Lamps [29], which show how OBJECT can be treated as an environmental and visualization factor.

### 4.1.1 Coffee Consumption Visualization from Bonfire

We describe Figure 5 of Bonfire; two bar charts detailing one's personal coffee consumption (the number of cups consumed and money spent) is shown only when a cup is placed on a table (Figure 3). Touching the visualization reveals more information on a laptop.

Using the framework, the user's intent is wanting to know about their personal coffee consumption. There is implicit interaction by showing and hiding the bar charts when a cup is placed and lifted from the table. Explicit interaction is achieved through touch input. For the environment factors, the OBJECT is table surfaces near a coffee cup; the LOCATION can be anywhere as tables are ubiquitous; and because tables are ubiquitous, PEOPLE'S privacy needs to be considered as strangers may be nearby in public locations. For the visualization factors, the DATA *Theme* is the money spent on coffee and the number of cups consumed, and the FORM is two simple bar charts with an *Anchor* to the cup. Green is associated with money spent, and red is used for the bar on cups consumed.

When we use our process to consider how this visualization was designed, we gain insight into some of these design choices. Because the LOCATION can be public and PEOPLE'S privacy needs to be considered in the environment stage, the authors possibly made other design decisions to protect privacy. For example, using simple bar charts as the FORM *Visual Encoding* shows the information at a very high *Level of Abstraction*, making it more appropriate to public settings. The implicit and explicit input allows the visualization to be shown at differing levels of detail, or not at all, giving the user some control over what is shown in different contexts.

Table 3: Example considerations during the design process, with a focus on SAR-related visualization considerations rather than general InfoVis or ubiquitous computing considerations.

| Consideration | Examples |
|---|---|
| Display on OBJECTS that are contextually relevant. | Display sleep data on bed; project on objects on/near dining table when eating breakfast. |
| Use FORM *Colours* and *Shapes* that blend into the environment by simulating natural material textures and colours. | Simulate patterns in wood grain. |
| Avoid projecting on OBJECT *Surfaces* that already serve another purpose, *Anchor* around the object. | TV screens, paintings. |
| Select contextually relevant and useful/interesting DATA. | Use data associated with daily updates during morning routine. |
| Adjust FORM'S *Level of Abstraction* based on PEOPLE'S *Activities*. | Use a high level of abstraction when the user is concentrating on another activity like exercising or driving. |
| FORM'S *Level of Abstraction* can be higher for visualizations that are lived with over long periods of time. | Users can associate colours with different types of tasks in a schedule visualization. |
| From privacy and collaboration standpoints, consider how the *Relationships* of the PEOPLE in the environment should influence the *Theme* of the DATA to display and which *Surfaces* to display on. | Hide sensitive information by using a higher level of abstraction when displaying a personal calendar; show relevant information on easily visible surface during a group meeting. |
| Make creative use of available *Surfaces* that could be associated with certain visual forms. | Banisters can be made to look like bar graphs. |
| *Which Surface* a visualization is projected on can also serve to encode information. | Use surfaces closer to the user to indicate urgency. |
| Avoid *Surfaces* or FORM *Anchorings* that are easily occluded by users when a visualization is being used. | Avoid projecting on the seat of a couch that is in use. |
| Select *Surfaces* that encourage users to take ergonomic postures. | Render text on surfaces perpendicular to the user's line of sight. |
| Use lighter FORM *Colours* in brighter environments to improve visibility. | Use pink rather than red when projecting on an outdoor banister. |

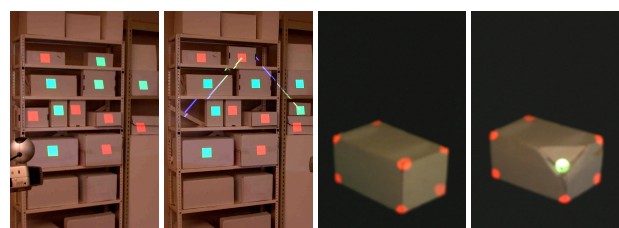

Figure 4: Warehouse box visualizations from RFIG Lamps (Figure 9 of Raskar et al. [29], cropped for detail).

### 4.1.2 Warehouse Box Visualization from RFIG Lamps

We describe the visualizations depicted in Figure 9 of the RFIG Lamps paper. In a warehouse, the visualizations serve the purpose of indicating changes in positions of boxes on storage racks and high-lighting deformation of boxes caused by moisture or heat. They do this by projecting coloured lines and shapes on the boxes (Figure 4).

In reference to the framework, the user's intent is to learn more about changes to the boxes, including box movement and possible damages and deformations. For the environment factors, the LO-CATION is inside a warehouse and the PEOPLE are the warehouse employees. For the visualization factors, the FORM uses blue-to-yellow gradient-coloured lines to indicate moved objects, and green and red circles indicating deformation and lack thereof, respectively; the DATA *Theme* comprises which objects have been moved and which parts of which objects have been deformed; and the OBJECTS are the storage boxes of interest.

Using our process as a lens, we can interpret the design decisions behind this RFIG Lamps visualization as follows. Because the LO-CATION is within a warehouse, the designers could assume that the PEOPLE would be warehouse employees. This would also suggest that the FORM could have a high *Level of Abstraction*, because the warehouse employees would be trained to interpret the visualization. The designers likely chose to display the visualization directly on the boxes because the DATA is about changes in boxes position and damages. In reference to FORM, the boxes already had red and green squares projected on them for a separate tagging purpose, so the designers likely chose to use a different set of colours (blue and yellow) to indicate box movement. The use of green and red circles to indicate deformation may be a creative decision that fits into the overall theme of saturated colours and abstract shapes.

## 4.2 Generating New Visualizations

We used the design framework and process to generate eighteen exemplar applications. As generating examples helped us iterate on the framework itself, creating eighteen applications allowed us to thoroughly explore different aspects of the framework. We focused on how SAR visualizations could enhance a typical day at home and the office. The exemplar applications are not intended to be novel by themselves, but instead function as demonstrations of how the framework could be used to create a wide variety of visualizations in different environments, and showcase the diversity of the space of SAR visualizations.

Within the HCI community, demonstrations are a valuable evaluation strategy that can show the breadth of examples covered by the dimensions and properties of a new design framework [19]. As our framework is theoretical, the exemplar applications allow us to evaluate the framework in a meaningful way without requiring a system implementation or a formal user evaluation. AR-CANVAS [1] used a similar approach, but with mock-up images rather than videos. Creating videos allows us to focus on the main concepts without technical restrictions introduced by a system implementation.

Visualizations were created using a variety of tools, primarily D3, p5.js, a free word cloud generator,[2] and Adobe Photoshop. In most cases, artificial data was used, as our focus is not on creating functional visualizations. To refine the appearance of the visualizations, we created mood boards using stills from the video footage to select colours that matched those of the room. Video footage was recorded at various home and office locations, and Adobe After Effects was used to superimpose the visualizations onto surfaces in the footage. We used the "Screen" blend mode to simulate how the visualizations could look with high-quality projectors.

For brevity, we only describe seven of the exemplar applications in detail, but summaries of all eighteen applications are given in Table 4. We selected these seven applications as they are more complex and cover a wide of range of environments, objects, and visual forms. Videos of all applications can be found in the supplementary resource. We recommend watching the corresponding conceptual video after reading each description to better understand the SAR visualizations (video names are shown in parentheses).

---

[2]https://tricklarnews.com/usa/cloudgenerator

Table 4: Summaries of all eighteen exemplar applications. Video titles marked with an asterisk (*) are only demonstrated in the supplementary materials. Videos of all applications are available in the supplementary materials.

| Video Title | Description |
| --- | --- |
| morning-sleep* | The headboard of a bed shows a line graph of historical sleep duration; the footboard shows coloured bands, representing the duration of sleep cycle stages. |
| morning-water* | A blue circular graph, highlighting water consumption, is shown around a faucet. |
| morning-breakfast | Summaries of social media updates and the news are placed around a phone and newspaper placed on a dining table; detailed views appear when these objects are lifted from the table. Weather summaries appear along the rim of a coffee cup. |
| morning-door* | Visualizations about someone's commute and their daily schedule appear on and around the front door; transit information is shown above the door, a speed comparison of transportation modes on a window, and a calendar on the door. |
| heading-car* | Visualizations about a driver's route and traffic appear when the car is off, but disappear when the car is turned on; ambient light indicating traffic congestion is shown on the steering wheel instead. |
| heading-office* | When someone enters an office lobby, abstract lines representing upcoming calendar events appear above a fireplace, but when they enter their personal office, a full calendar view is shown on their desk. |
| office-cabinet | Pictographs highlighting objects placed inside a cabinet are shown on the drawers; the images change to abstract lines if strangers enter the room. |
| office-coffee* | Depending on the TIME, different visualizations are shown around the coffee machine of a common lounge. If the person is in the middle of a meeting, a line graph about company revenue is shown; otherwise sports data is shown. |
| lunch-spending* | Abstract bar charts showing spending activity are shown on balusters and stair risers outside a restaurant. |
| lunch-schedule | Different visualizations show how busy someone's schedule is as they move to different environments; the balusters outside a restaurant show bar charts, a circular table shows a pie chart, and a window sill shows heatmap. |
| office-reminders | A graph moves closer to someone and becomes more opaque as a meeting time approaches; it can be dismissed using a voice command. |
| office-meeting | When a meeting organizer is alone, a timeline of personal deadlines appear along a nearby table edge, but when a meeting attendee enters the room, the timeline moves to a flipchart and shows information about deadlines relevant to both people. |
| memories-plant | An infographic placed on a potted plant shows how much it has grown; labels appear with explicit touch input. |
| memories-art* | Infographics placed around art pieces show information about when and where each piece was purchased, the estimated worth over time, and key events in the artist's life. |
| memories-couch* | A timeline of old photos featuring a couch are shown along its seam. |
| memories-chair* | Bar charts showing who has used a dining chair the most appear along the wooden spindles; as people sit on the chair, they leave virtual markings on the seat. |
| dashboard-tv | Recent viewing activity, recommendations, and average television watch time are shown on surfaces around the television, as the television is already showing other content. |
| dashboard-wifi* | Bandwidth information is placed near a router and a wireless printer shows the number of pages that connected devices have printed. |

### 4.2.1 Daily Updates on a Dining Table (morning-breakfast)

Visualizations placed around different objects on a dining table can be used to convey information about daily updates, such as the news, weather, and social media updates (Figure 5). If a newspaper is placed on the table, a bar chart appears along the edge, summarizing the number of news stories by category. When the user begins reading it, a larger word cloud about its contents appears on the table. Similarly, if a phone is placed on the table, a bar chart summarizing the number of unseen social media posts by platform is shown along the edge. When the user begins browsing through their social media profile, trending hashtags appear on the table as a series of line graphs. Finally, a "weather clock" appears along the rim of a coffee cup. The circumference represents a 24-hour day, and each colour represents a different weather condition. In reference to the framework, the User Intent is to view information about daily updates while seated at the dining table for breakfast. Information typically accessed at this TIME of day informs the DATA *Themes* (news, social media, and weather updates). News and social media data suggests that OBJECTS commonly placed on the table should be used for the visualization (newspaper and phone). Both OBJECTS may be lifted off the table and used, so the FORM must adapt to these changes in OBJECT position (expanded view on the table). A coffee

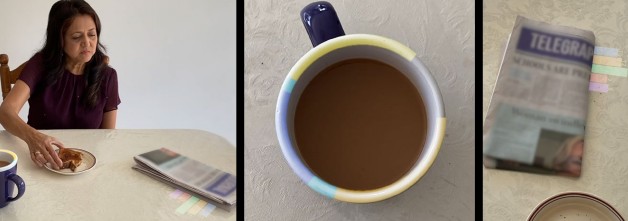

Figure 5: News, Weather, and Social Media Updates at the Breakfast Table (morning-breakfast)

cup is another commonly available OBJECT that could be used to show weather DATA in an embedded way and its *Shape* informs the design of the FORM (colours along the circular rim).

### 4.2.2 Pictograph of Objects Inside a Cabinet (office-cabinet)

Pictographs placed on a cabinet can be used to relay information about stored items (Figure 6). Each item placed inside the cabinet is represented using a different icon. Its location represents the drawer in which the item is placed, and the number of icons shows how many of each item is placed inside that specific drawer. The colour

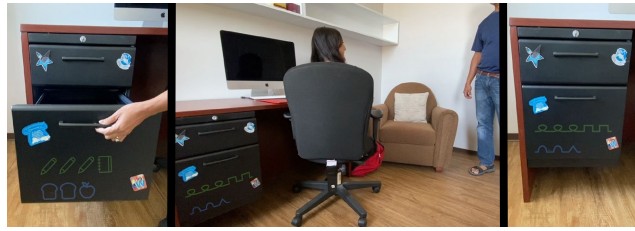

Figure 6: Pictograph of Objects Inside a Cabinet (`office-cabinet`)

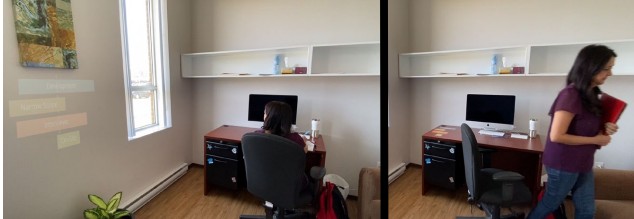

Figure 8: Visualizations as Meeting Reminders (`office-reminders`)

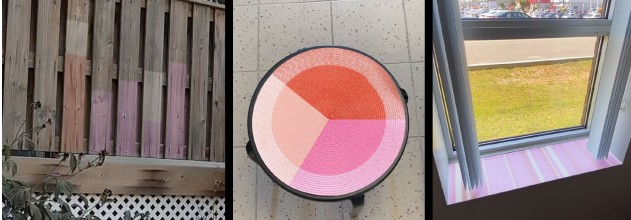

Figure 7: Schedule for the Afternoon Outside a Restaurant and in the Office (`lunch-schedule`)

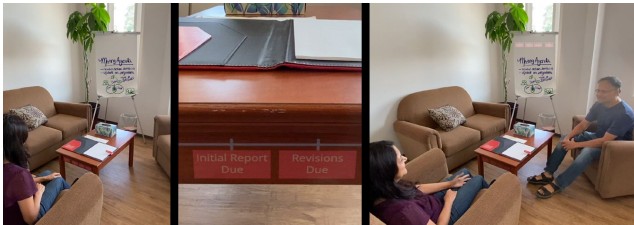

Figure 9: Personal and Shared Visualizations in a Meeting (`office-meeting`)

encodes the type of object (e.g. office supplies and snacks). When someone else approaches the cabinet, if they do not have a close relationship, the visualization changes to protect the person's privacy where abstract line patterns are used to convey the same information. Each item is represented using a unique pattern, rather than an icon. This approach is similar to past work on privacy-preserving large displays, which can change the level of detail of information shown based on the people present [32, 33].

In reference to the framework, the *Function* of an OBJECT in the room (office cabinet used for holding items) and the PERSON'S *Activity* (working in private versus working with another person present) and *Relationship* to other people (whether the other person is a stranger) are used to inform the DATA *Theme* (objects stored) and the FORM'S *Level of Abstraction* (pictographs versus line patterns).

### 4.2.3 Schedule Outside and in the Office (`lunch-schedule`)

Information about a person's schedule is shown on a banister outside. Each pink bar represents how busy the person is for each remaining hour of the workday. White bars show how busy they are on average, and only appear if they are less busy than usual. Red highlights the current hour. When they enter their office, a pie chart appears on top of a circular table, showing them how busy they are for the remainder of the day. Each colour represents a different task. Finally, when they enter their personal office, they can see a heatmap of their remaining tasks along the window sill. Each colour along the heatmap represents different types of tasks over time (Figure 7).

In reference to the framework, the DATA *Theme* (relative proportions of time) informs the FORM *Visual Encoding* (bar chart), and the surface *Shape* being projected on (banister, circular table) informs the style of the visualization (chart type, colours, and bar orientation).

### 4.2.4 Visualizations as Meeting Reminders (`office-reminders`)

Visualizations could be used to gently remind someone of an upcoming meeting (Figure 8), similar to how ambient displays have been used to remind users of upcoming tasks [23] and to take breaks [13]. Over time, the visualization gets closer to the person and becomes less transparent, as a way to inform the user by emphasizing objects that are nearby [34]. When someone is working at a desk and they have a meeting in 30 minutes, data that is relevant to their upcoming meeting, such as a Gantt chart, is shown on a nearby wall. Because there are still 30 minutes, it is placed further away from the person and with 30% transparency. After 15 minutes, the Gantt chart fades

away, and it reappears with 50% transparency on another wall that is closer to and facing the person. Finally, when there are only 5 minutes before the meeting, the Gantt chart moves to the top of their desk and is now opaque. This indicates that the person should begin packing up their belongings in preparation for the upcoming meeting. At any point, they can issue the voice command "remind me later" to dismiss the visualization until shortly before the meeting.

In reference to the framework, the TIME (relative to the next meeting) is used as contextual information to inform the OBJECT *Surface* being projected on (from a far surface, the wall, to a close surface, the desk) and the FORM'S *Transparency*.

### 4.2.5 Personal and Shared Views (`office-meeting`)

Visualizations can change their position and underlying data to be more visible and useful to the people in the room. Before a meeting, the organizer may want to review their personal deadlines for a project before other attendees arrive. When they are alone, a timeline view of deadlines that are relevant to them are placed in front of them, along the table edge, in a similar way to work by Joshi and Vogel [17]. When other attendees arrive for the meeting, the timeline previously on the table edge moves to a surface that is more convenient for everyone to view, a flip chart. A filter is applied to the data so the project deadlines shown are relevant to the entire group, not just the meeting organizer. After the meeting, the timeline moves back to the table edge in front of the organizer and displays personal deadlines, allowing them to review their deadlines once again (Figure 9).

In reference to the framework, the User Intent is to view personal information about a project schedule before and after a meeting, but shared information during the meeting. The DATA *Theme* (project schedule) informs the FORM *Visual Encoding* (timeline). The DATA *Theme* and multiple DATA *Owners* (personal or shared data) suggests that a DATA *Filter* is needed, to be applied as people enter and exit the room. The different DATA *Owners* also suggests that multiple OBJECTS will be needed: personal information implies that the content should be closer to the user [34], and the availability of a table surface in front of the organizer suggests it should be used as a personal display. The flip chart is close to the group and is a good candidate for a shared display.

### 4.2.6 Plant Infographic (`memories-plant`)

An infographic placed on a potted plant can show how much it has grown. Images representing different growth states, like flowering,

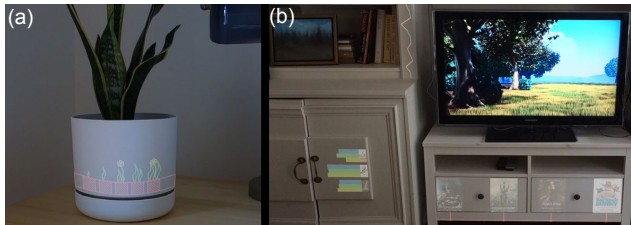

Figure 10: (a) Plant Infographic (`memories-plant`); (b) "Dashboard" for TV Data (`dashboard-tv`)

are placed around the pot (Figure 10a). An infographic placed on the pot is arguably more aesthetically appealing than other visualizations, and users may wish to preserve the artistic integrity when the information is not actively being sought out, similar to informative art [31]. As such, additional information, like labels showing the exact height of the plant, are only revealed when they tap on the infographic.

In reference to the framework, the User Intent is to see an infographic on their potted plant, informing the FORM *Visual Encoding* (infographic) and OBJECT (the plant's pot). The OBJECT informs the DATA *Theme* (growth state data), which informs the FORM *Colours* (green for plant data). The FORM *Visual Encoding* suggests a creative decision of revealing detailed information with user input, suggesting explicit User Interaction (tapping the pot).

### 4.2.7 "Dashboard" for Television Data (`dashboard-tv`)

The surfaces around a television can be used to create a dashboard about one's television consumption (Figure 10b). Recent viewing activity is shown as a timeline on a television stand. Each point represents when something new was viewed, and a promotional image for the media is shown below. Recommendations are placed on a nearby cabinet. Promotional images are placed to the right, and a bar graph showing critic scores are placed to the left of the image. Finally, a line graph placed on the edge of a bookshelf shows how much television has been viewed on average.

In reference to the framework, the User Intent is to see information about their television consumption. The DATA *Theme* (television consumption) informs the OBJECT (television), but the television is not a good surface for projection as it can also display information, and the SAR and television displays may conflict. This suggests the FORM'S *Anchor* should be surfaces around the television instead.

## 5 DISCUSSION AND FUTURE WORK

Our design framework serves as a foundation for future work on SAR visualizations. We discuss example applications and extensions.

### 5.1 Privacy Considerations

SAR-based information visualization systems raise a number of potential privacy concerns. Because SAR enables multiple people to view a visualization, it is susceptible to *shoulder-surfing*, or onlookers such as strangers observing personal information. If sensitive data is visualized, the user may want to ensure that only they are authorized to observe and use the visualization. This concern of shoulder-surfing is shared with large displays; we suggest that SAR visualizations can learn from how this threat is mitigated in that domain. For example, Brudy et al. [4] provide awareness of onlookers through visual cues, and the display moves items out of an onlooker's view, or blacks them out, to help preserve privacy. In a similar vein, some of our exemplar visualizations have dynamic levels of abstraction based on the context, like `office-cabinet`, which uses abstract shapes to represent cabinet contents when strangers are present, or `heading-office`, when the user's calendar is only represented as an abstract heatmap that will not convey as much sensitive information to others. Another way to mitigate shoulder-surfing would be to use

a combined approach with public or shared content projected with SAR, and sensitive content displayed on personal HMDs, similar in implementation to Benko et al.'s FoveAR system [2].

### 5.2 Technical Feasibility

We believe many of the exemplar applications would be feasible using existing technology. Computer vision, IoT devices, and other hardware sensors could be used to identify people and objects in the environment, giving insight into who is at a specific location and what they are doing. It may be difficult for hardware to determine the relationships between people, for example, to ensure that strangers cannot see one's personal data. Instead, relationships could be inferred using other forms of context, such as from social media profiles or personal calendars. To place content in the environment, existing toolkits for multi-projector SAR setups, such as RoomAlive [15], could be used. Projecting outdoors may be a challenge, but systems like Lightform [20] enable high-quality projections, even in bright and outdoor environments.

When considering privacy, SAR generally requires sensors and stationary projectors. This infrastructure would need to be protected from hacking and misuse of user data. If shared infrastructures are not regulated, personal information could be used for purposes other than for information visualization, commonly referred to as "secondary use." For example, it could be used for advertising, or connection information could be used as a metric to track people's movement. One solution is to build SAR infrastructures around notions of "zero-knowledge" services. For example, visualizations could be pre-rendered on client devices, and protocols between centralized systems and user devices could be designed to not expose any unique identifiers. Another approach would be to use portable SAR systems, like to OmniTouch [10] or AAR [11]; people would carry around portable projectors for public or shared SAR visualizations, without relaying sensitive information to centralized servers.

We also envision future systems that could create SAR visualizations automatically, sensing and using information from the environment through heuristics, lookup tables, or more sophisticated approaches like machine learning. Data and objects on which to visualize could be selected based on conceptual relationships between them, to enable creating relevant visualizations. A system could automatically extract dominant or accent colours from nearby surfaces and create a "colour palette" for a visualization [26], making it more aesthetically pleasing or less obtrusive.

### 5.3 Extending to Other Forms of AR

We focused on SAR because it presents specific challenges and opportunities like privacy concerns and shared visualizations. However, our framework and methodology may be more generally applicable and extensible to conventional AR forms like HMD-based AR, because the factors in our framework draw from work on situated and embedded visualizations, context-aware computing, and InfoVis design frameworks. For example, similar to an "electronic post-it note" [34], a visualization could show personal reminders in an HMD based on the TIME (e.g. "go to the dentist at 10:00"), the LOCATION of the user or OBJECTS at their location (e.g. "remember to clean the coffee filter when entering the kitchen"), or the PEOPLE in the vicinity (e.g. "remember to give the book to Bob"). The FORM might vary in terms of *Colour* and *Level of Abstraction* to minimize distraction depending on the LOCATION, and the Interaction (e.g. to dismiss, share, or learn more about a reminder) could vary depending on the affordances available at the LOCATION.

A noteworthy aspect is that our SAR framework differs from HMD-based AR because of a primary emphasis on *objects*. Our framework places OBJECT at the centre of the Venn diagram, signifying its importance both from a standpoint of context as objects in the environment, and a standpoint of visualization as a collection of surfaces on which to display. In contrast, for HMD-based AR, con-

tent can be placed in mid-air, detached from the surface of specific objects. The framework could be extended to consider ways that visualizations could be rendered off the surfaces of objects, as well as the notion of simultaneous public plus private views.

### 5.4 Formal Evaluations

Using our exemplar applications as a starting point, expert and user evaluations could be conducted to further enhance and validate our framework. For example, experts could participate in design workshops to create SAR visualizations for use in additional contexts. To better understand the technical feasibility and usability of SAR information visualizations enabled by our framework and associated design process, our exemplar applications could be implemented as hardware prototypes and evaluated with people with different demographics and abilities. Specific aspects of each visualization, such as the readability of colours and text, the level of information retention, and the level of distraction, could be compared to visualizations on computer screens or in conventional AR.

## 6 Conclusions

We contribute a design framework for embedded SAR visualizations that capture interdependent relationships between contextual information and existing objects in an environment. We demonstrate how it can be used to create a variety of visualizations over the course of the day through a series of exemplar applications using envisionment videos. Our work serves as a foundation for designers interested in creating visualizations that better integrate into daily life.

## 7 Acknowledgements

We thank Dr. Arif Bungash for allowing us to film at his office. We also thank Dr. Vinod Joshi and Radhika Joshi for acting in the envisionment videos. This work was made possible by the NSERC Discovery Grant 2018-05187, the Canada Foundation for Innovation Infrastructure Fund 33151 "Facility for Fully Interactive Physio-digital Spaces," and Ontario Early Researcher Award ER16-12-184.

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
