# OpenReview forum: "A Design Framework for Contextual and Embedded Information Visualizations in Spatial Augmented Reality"
_graphicsinterface.org/Graphics_Interface/2022/Conference — GI 2022_

### Official Review · Reviewer_Gius · 2021-12-28
**The paper presents a significant work in the field of embedded information visualization in SAR space. It proposes a design framework that focusses on relationship between environment and the visualization.**

**Rating:** 8
**Confidence:** 5

**Review:**

Quality: of the paper is excellent. The authors have discussed existing frameworks and identified a crucial gap in visualising information in SAR. The paper also provides ample examples.
Clarity: There are minimal grammar errors. the content is easy to understand. Please see the comments below for some questions that I had in mind, the authors could think of clarifying this in the paper.
Originality: The topic is original and is very important for the field of HCI and graphic visualization in AR space.
Significance: the framework is interesting, and worth evaluating in different environments and AR types (HMD?)

Feedback:
(1) The paper should discuss the transferability of this framework to other types of AR such as head mounted. One primary difference between SAR and other types is the ability of SAR to allow multiple people interact with the AR space and the elements. Considering that, people - people relationships and context is critical to consider within the environment. How would this affect the visualization? Also, the examples shown do not consider all the elements of the environment - people and objects. For example, a cup without the user does not give the context. so the relationship between object and people is also important.
(2) The paper could explain future work and limitations of the framework. One of the areas of future work could be to evaluate it with people from different demographics and with different abilities. Accessibility of the information visualtion is critical if the framework criteria is to connect environment with information.

---

### Official Review · Reviewer_p4Qi · 2022-01-08
**This is a nice design study/model paper that is focused on an emerging aspect of AR, but it needs a bit of work.**

**Rating:** 8
**Confidence:** 4

**Review:**

This paper presents a design framework for spatial augmented reality. It includes an analysis of two existing works and a set of mock-up designs, each of which are described in the context of this framework.

This paper is a combination of a "design study" and "model" (following Munzner's classification of InfoVis papers: https://doi.org/10.1007/978-3-540-70956-5_6). I suggest commenting on this classification in the introduction, and highlighting the benefit of such a style of research. After doing so, the authors may want to re-visit the second paragraph of Section 4.2 to decide whether that justification is needed anymore or whether it should be integrated into the discussion on the design study/model contribution of this paper.

The framework itself is a novel extension of other visualization frameworks, and is well explained. The background work is well researched and presented. While Figure 2 is a good overview of the relationship between the user intent, environment, and visualization, I think some careful attention needs to be given to the arrows (pointing to influenced factors).  For example, the figure shows the user's intent influencing the environment, which doesn't make sense to me since one's intent is solely a cognitive aspect. Please check these details carefully to ensure that they accurately represent the model.

One of the main contributions of this work is with respect to the environment aspect of the model. As such, Section 3.3.2 is a critical section of this paper. As it stands, this is not explained as clearly as the other aspects (user intent and visualization).

I like that the authors have used the framework to both explain the work of others and to present their own design studies/scenarios that follow the framework. Regarding the new scenarios, I can see the point of wanting to provide a complete "day in the life" showing how SAR-based visualizations can be used. However, the space constraints of the paper have resulted in just seven of the 18 scenarios being explained. This seems to me to be a bit of a back-door for exceeding the page limit of the paper. I would like to see further discussion about why these seven are provided (it seems to me that they each illustrate something different in terms of the influence of the environmental elements among themselves, and how this affects the visualization). Perhaps small-multiples versions of Figure 2 can be added to the scenarios that illustrate what is unique in each. For each scenario, the authors may want to consider putting the explanation of the framework in separate paragraphs, so that this important aspect stands out from the explanation of the scenario.

Overall, I like this paper. It shows that the authors have given a lot of thought to the possibilities for SAR, and have come up with a framework that inserts the environment within the typical user intent and visualization concerns. While some may argue for the need for fully functioning implementations, these can come later. This paper provides a vision for the future of SAR visualizations, and establishes some terminology that can be used to describe and compare different approaches in the future.


Other Comments:

1. I don't agree with the statement that SAR "avoids the privacy concern of continuously sensing the environment using a camera", as some such systems may use cameras and other sensors to monitor the space.

2. The Venn diagram is mentioned in the introduction, but not presented until later.

3. In section 4.2, there is a comment "our contribution is theoretical", which I do not think is accurate. See my comment above about Munzner's classification of InfoVis papers.

4. Please check the reference format and style carefully. There are a number of minor errors that should be corrected.

---

### Official Review · Reviewer_JGQj · 2022-01-15
**Well-written, long, hard to judge the value**

**Rating:** 6
**Confidence:** 2

**Review:**

The paper is very well written; however, it is a bit verbose (and thus, at 12 pages, is rather long). The same can be said about the accompanying video.

The authors propose a framework targeting visualization in Spatial Augmented Reality (SAR).
Although they do mention several factors distinguishing SAR from, say, HMD-based AR, and do mention that their framework extends some of the existing models, I think the differences are not articulated clearly enough. As far as I can tell, the only distinguishing aspect of SAR discussed in the paper is the issue of privacy, and other than that, the interactions described are not impossible with other types of augmented reality.

While frameworks are clearly important for design, I think the value of the presented framework has not been demonstrated clearly enough. E.g., what aspects of design would not have been possible with the other models. It appears that one needs to peruse this paper, as well as the dozens of citations at the end, to confirm the similarities and the differences between the presented framework and others, like AR-CANVAS.

Pros
the paper is written in a clear language; has valuable, albeit small, illustrations.

Cons
the paper is unnecessarily lengthy, in my opinion. The differences between the presented framework and other frameworks are not articulated clearly enough.

---

### Decision · Program_Chairs · 2022-01-18

Accept